# Hydrogen Bond Assisted Three-Component Tandem Reactions to Access *N*-Alkyl-4-Quinolones

**DOI:** 10.3390/molecules28052304

**Published:** 2023-03-02

**Authors:** Huanhuan Liu, Huadan Liu, Enhua Wang, Liangqun Li, Zhongsheng Luo, Jiafu Cao, Jialin Chen, Lishou Yang, Xiaosheng Yang

**Affiliations:** 1State Key Laboratory of Functions and Applications of Medicinal Plants, Guizhou Medical University, Guiyang 550014, China; 2The Key Laboratory of Chemistry for Natural Products of Guizhou Province and Chinese Academy of Sciences, Guiyang 550014, China; 3Department of Food and Medicine, Guizhou Vocational College of Agriculture, Qingzhen 551400, China

**Keywords:** hydrogen bond, polyphosphoric acid, polyphosphate ester, *N*-alkyl-4-quinolones, neuroprotective activity

## Abstract

Hydrogen-bonding catalytic reactions have gained great interest. Herein, a hydrogen-bond-assisted three-component tandem reaction for the efficient synthesis of *N*-alkyl-4-quinolones is described. This novel strategy features the first proof of polyphosphate ester (PPE) as a dual hydrogen-bonding catalyst and the use of readily available starting materials for the preparation of *N*-alkyl-4-quinolones. The method provides a diversity of *N*-alkyl-4-quinolones in moderate to good yields. The compound **4h** demonstrated good neuroprotective activity against *N*-methyl-ᴅ-aspartate (NMDA)-induced excitotoxicity in PC12 cells.

## 1. Introduction

A valuable class of nitrogen-containing heterocycles is 4-Quinolone, which widely exists in natural products [1,2], synthetic blocks [3,4,5,6], and bioactive molecules [7,8,9,10,11,12,13]. Typical *N*-alkyl-4-quinolone derivatives are drugs such as ciprofloxacin, norfloxacin, and oxolinic acid, which have emerged as potent antibiotics (Figure 1). Traditional synthetic approaches to the 4-quinolone framework are based on cyclocondensation involving Conrad–Limpach [14,15,16,17], Niementowski [18,19,20,21] and Camps-type reactions [22,23,24], which usually suffer from harsh reaction conditions, limitation of substrate scope, and the requirement of commercially unavailable starting materials. Recently, several elegant procedures have focused on the use of transition-metal catalysis [23,24,25,26,27,28,29,30,31] and tandem reactions from *o*-alkynylbenzamides/aldehydes or *o*-haloaryl acetylenic ketones/amines (Figure 1a) [32,33,34,35,36]. However, these successful routes require transition-metal catalysis, multistep procedures, long reaction time, and special precursors such as prehaloaryl acetylenic ketones. Additionally, most of these methods failed to give *N*-alkyl-substituted 4-quinolones, that are featured in many clinically used drugs. Therefore, the implementation of mechanistically different transformations to achieve structural diversity in 4-quinolone synthesis from easily accessible raw materials still remains a large need.

Hydrogen-bonding interactions play a crucial role in catalysis, especially in the field of asymmetric catalysis [37]. Organic chemists have found that small molecules possessing distinct hydrogen bond donors, such as O-H, N-H, and S-H functional groups, catalyze a range of C-C and C-heteroatom bond-forming reactions, although the hydrogen-bonding interactions are weak [38,39,40,41,42,43]. To date, hydrogen-bonding catalysts possess a wide range of functional and structural frameworks are explored, including thioureas [44,45], prolines [46,47], sulfonamides [48] and chiral phosphoric acid (CPA) [49,50,51,52,53,54,55,56,57].

Investigations of CPA in the field of hydrogen-bonding catalytic chemical reactions made clear that the P=O and O-H serve as hydrogen bond acceptor and donor, respectively [49,50,51,52,53,54,55,56,57]. Previous reports show that polyphosphate ester (PPE) could activate the nitrogen functional group toward nucleophilic attack, and serve as alkylating reagents [58]. Recently, our group reported polyphosphoric acid (PPA) promoted tandem reactions for the synthesis of heterocycles, which revealed that PPA is an effective condensation reagent [59,60]. On the basis of the previous work, we envisioned that *N*-alkyl-4-quinolones could be synthesized via the three-component tandem reactions of readily accessible 2-aminoacetophenones, aldehydes and alcohols using PPA (Figure 1b). Specifically, PPE **6** and condensation product **7** were formed in the presence of PPA, and subsequent formation of PPE-**7** complex **8** via hydrogen-bonding interactions, which are the driving force for the formation of **9**. Immediately, **9** undergoes alkylation to produce intermediate **10** followed by a tautomerization to deliver the 4-quinolone **4y**.

## 2. Results and Discussion

To support this hypothesis, several control experiments and density functional theory (DFT) calculations were performed (Figure 2). Fortunately, the desired product **4y** could be obtained directly from **1a**, **2a** and **3c** in 37% yield (Figure 2a). As depicted in Figure 2b(1), condensation product **7** was afforded in the 32% isolated yield. Subsequently, **7** was also employed as substrate and the 4-quinolone **4y** was obtained in 35% yield (Figure 2b(2)). Furthermore, PPE **6** was detected by ^31^P NMR (Figure 2b(3), see Appendix A for details). As shown in Figure 2c, diphosphoric acid ethyl ester **11** (a model PPE) reacts with **7** to produce complex **12** with a computed free energy of 1.6 kcal/mol. Subsequently, the intermediate **13** was generated via transition state **TS1**, which then undergoes *N*-alkylation to yield product **4y**. The DFT calculations suggest that this process is thermodynamically feasible. Additionally, the generation of such a PPE-**7** complex **8** was further verified by ^31^P NMR (see Appendix A for details).

Next, we commenced our studies using 2-aminoacetophenone **1a**, benzaldehyde **2a** and 1-octanol **3a** as model substrates to optimize the reaction conditions. A mixture of **1a**, **2a**, **3a** and PPA (5.0 equiv.) in DMF refluxed for 1 h under nitrogen atmosphere to give the target compound **4a** in 38% yield (Appendix A, entry 1). PPA amount, time and temperature screening revealed that 1.0 equiv., 3 h and reflux was the best choice (see Appendix A for more details). We further screened the additives including P_2_O_5_ and PPA/P_2_O_5_ (Table 1, entries 1–2). According to the screening results, PPA was identified as the best. The effect of solvent was also surveyed, yet the results were inferior to those of DMF (Table 1, entries 3–8 vs. Appendix A, entry 7). Subsequently, reaction was performed in the presence of 4 Å MS indicated that 4 Å MS had no effect on this transformation (Table 1, entry 9). Finally, the optimized reaction conditions were determined as follows: **1a** (1 equiv.), **2a** (1.2 equiv.), **3a** (1 mL) and PPA (1.0 equiv.) in DMF refluxed for 3 h under a nitrogen atmosphere (Appendix A, entry 7). Under the optimized conditions, a gram scale reaction was conducted and provided **4a** in 70% yield (Table 1, entry 10).

With the optimized reaction conditions in hand, we then explored the scope and limitations of this reaction (Table 2). Gratifyingly, diverse 2-aminoacetophenones, aldehydes and alcohols were compatible with the reaction, producing the target 4-quinolones 4a-4ao in satisfactory yields. As can be seen, the electron-donating group (methyl or ethyl) on benzaldehyde led to higher yields, while the electron-withdrawing group (F, Cl or Br) gave lower yields (**4a**, **4e** and **4f** vs. **4b**–**4d**). It is noteworthy that naphthaldehyde, thenaldehyde and cyclohexanecarboxaldehyde also underwent smooth transformation to give corresponding 4-quinolones (**4g**–**4i**). Then, the scope of the reaction was evaluated regarding various aminoacetophenones with different substituents, and it was found that electron-withdrawing groups, such as F, Cl and Br, provided lower yield (**4j**–**4l** and **4n**–**4p** vs. **4m** and **4q**). We further examined the scope of alcohol substrates. Different alcohols worked well with aminoacetophenones and aldehydes yielding diverse *N*-alkyl-substituted 4-quinolones **4r**–**4ao** in moderate to good yields (48–84%). It was also observed that this reaction was sensitive to the steric hindrance of the alcohols. The yield was decreased with increasing the steric hindrance (e.g., **4r** vs. **4y** vs. **4af** vs. **4am** vs. **4an** vs. **4ao**). Next, we carried out a reaction between 2-aminoacetophenone, benzaldehyde and phenol. Unfortunately, phenol was not tolerated in this case, and the target product **4ap** was not detected.

Interestingly, phenols were compatible with the reaction in the presence of Pd/C (Table 3, optimization study see Appendix A for more details). A π-alkene–palladium complex might be formed [61], which would promote the cyclization of intermediate **8**. However, the use of palladium catalysts proved to be less effective in the case of alcohols (Appendix A, entries 12–14 and Appendix A, entry 18).

In order to gain deeper insight into this reaction, phosphates (**14** and **15**) were subjected to the reaction, giving the target products in 29% and 36% yields, respectively (Figure 3a). The electron-withdrawing F substituent has long been proposed to act as a nonclassical hydrogen bond acceptor on the aryl ring [62,63]. We then introduced a hydrogen-bonding acceptor to 2-aminochalcone; **16**, featuring the electron-withdrawing F, was tested. The **16** presented product **4u** with a better yield then that of **4r** (Figure 3b vs. Figure 3c). *N*-methyl substituted chalcone **17** provided the desired product **4r** in 2% yield, probably owing to the fact that hydrogen-bonding interactions were weakened by the methyl (Figure 3d). These observations collectively suggested that the hydrogen bonds between PPE and 2-aminochalcones might be formed.

Previous studies show that quinolones were identified as potent NMDA/glycine antagonists with neuroprotective properties [58,64,65]. Thus, all the 4-quinolones were evaluated for their neuroprotective activity against NMDA-induced injury in PC12 cells. The substituent on the nitrogen of 4-quinolone had effect on the activity. *N*-octyl -substituted compounds exhibited better activity than other targeted compounds (Appendix A). Among the compounds, the most active is compound **4h**, exhibiting neuroprotective potency similar to the MK-801 at 20 μM (Figure 2). To investigate the possible binding mode for this series, **4h** was docked into the NMDA receptor. In the binding mode, **4h** fitted well in the active pocket. The binding energies of **4h** and DCKA were −7.46 kcal/mol and −7.65 kcal/mol, respectively (see Appendix A for more details).

## 3. Materials and Methods

### 3.1. General Information

Fetal bovine serum, Dulbecco’s modified Eagle’s medium, penicillin and streptomycin were obtained from Gibco (Gibco, Paisley, UK). 3-(4,5-Dimethylthiazol-2-yl)-2,5-diphenyl-tetrazolium bromide (MTT) and NMDA were purchased from Sigma-Aldrich (Saint Louis, MO, USA). MK-801 was from MCE (Shanghai, China). PC12 cells from Institute of Cell Biology (Chinese Academy of Sciences, Shanghai, China). Unless otherwise noted, all reagents, catalysts and solvents were purchased from commercial suppliers and used without further purification unless otherwise noted. Column Chromatography was performed with silica gel (200–300 mesh). Melting points were determined using an X-4 melting point apparatus with microscope. The IR spectra were recorded with a Mattson FTIR spectrometer 5000. Absorption maxima were measured in cm^−1^. ^1^H NMR (600 MHz) and ^13^C NMR (151 MHz) spectra were achieved in CDCl_3_ on a Bruker AVANCE 600 MHz spectrometer. High-resolution mass spectra were measured on a ThermoFish QE Focus facility (Waltham, MA, USA).

### 3.2. Synthesis of Compounds **4a**–**4ao**

General procedure. To a solution of DMF (0.5 mL) was added 2-aminoacetophenones **1** (0.37 mmol), benzaldehydes **2** (0.44 mmol), alcohols **3** (1 mL) and PPA (0.37 mmol) in a 50 mL round bottom flask. The reaction mixture was refluxed for 3 h. The solution was quenched with water and the organic layer was dried over Na_2_SO_4_, filtered and evaporated. The resulting crude compound was purified by silica gel column chromatography using hexane/ethyl acetate mixtures to afford the corresponding products **4a**–**4ao**.

*1-Octyl-2-phenylquinolin-4(1H)-one* (**4a**) [32]. Yellow solid; Yield: 81%; Mp: 58.8–61.5 °C; IR (KBr plate) ν_max_ 2945, 2920, 1617, 1594, 1358, 835, 694. ^1^H NMR (600 MHz, CDCl_3_) δ 8.26–8.24(dd, *J* = 8.3, 1.5 Hz, 1H, CH Ar), 8.13–8.13 (m, 3H, CH Ar), 7.75–7.72 (ddd, *J* = 8.4, 6.8, 1.5 Hz, 1H, CH Ar), 7.56–7.54 (dd, *J* = 8.3, 6.9 Hz, 2H, CH Ar), 7.52–7.47 (m, 2H, CH Ar), 7.19 (s, 1H, CH), 4.30–4.28 (t, *J* = 6.4 Hz, 2H, CH_2_), 2.03–1.98 (dt, *J* = 15.0, 6.6 Hz, 2H, CH_2_), 1.64–1.59 (m, 2H, CH_2_),1.48–1.43 (m, 2H, CH_2_), 1.42–1.38 (m, 2H, CH_2_), 1.37–1.33 (tq, *J* = 7.2, 3.9, 3.0 Hz, 4H, CH_2_), 0.95–0.93 (t, *J* = 6.8 Hz, 3H, CH_3_). ^13^C NMR (151 MHz, CDCl_3_) δ 162.29 (CO), 158.89, 149.25, 140.52, 129.91, 129.20, 128.75, 127.61, 125.26, 121.77, 120.56, 98.60, 68.47, 31.85, 29.36, 29.27, 28.99, 26.17, 22.70, 14.14. HRMS Exact mass calcd. for C_23_H_28_ON [M + H]^+^: 334.21654; found: 334.21594.

*2-(4-Fluorophenyl)-1-octylquinolin-4(1H)-one* (**4b**). Yellow solid; Yield: 77%; Mp: 47.6–57.1 °C; IR (KBr plate) ν_max_ 2920, 2850, 1610, 1590, 1353, 824, 626. ^1^H NMR (600 MHz, CDCl_3_) δ 8.25–8.23 (dd, *J* = 8.3, 1.5 Hz, 1H, CH Ar), 8.14–8.09 (m, 3H, CH Ar), 7.74–7.71 (ddd, *J* = 8.4, 6.8, 1.5 Hz, 1H, CH Ar), 7.52–7.49 (ddd, *J* = 8.1, 6.8, 1.2 Hz, 1H, 1 CH Ar), 7.24–7.20 (m, 2H, CH Ar), 7.13 (s, 1H, CH), 4.30–4.28 (t, *J* = 6.4 Hz, 2H, CH_2_), 2.03–1.98 (m, 2H, CH_2_), 1.62–1.58 (p, *J* = 7.4 Hz, 2H, CH_2_), 1.48–1.43 (dq, *J* = 9.2, 6.6 Hz, 2H, CH_2_), 1.41–1.37 (m, 2H, CH_2_), 1.35–1.33 (qd, *J* = 6.7, 2.7 Hz, 4H, CH_2_), 0.94–0.92 (t, *J* = 6.9 Hz, 3H, CH_3_).^13^C NMR (151 MHz, CDCl_3_) δ 163.72 (q, ^1^*J*_CF_ = 249.15), 162.40 (CO), 157.72, 149.17, 136.60, 136.58, 130.03, 129.41 (q, ^3^*J*_CF_ = 9.06), 129.08, 125.33, 121.78, 120.46, 115.64 (q, ^2^*J*_CF_ = 25.67), 98.22, 68.51, 31.83, 29.35, 29.25, 28.98, 26.16, 22.68, 14.13. HRMS Exact mass calcd. for C_23_H_27_ONF [M + H]^+^: 352.20712; found: 352.20651.

*2-(4-Chlorophenyl)-1-octylquinolin-4(1H)-one* (**4c**). Yellow solid; Yield: 78%; Mp: 62.1–65.1 °C; IR (KBr plate) ν_max_ 2924, 2848, 1617, 1593, 1352, 818, 668. ^1^H NMR (600 MHz, CDCl_3_) δ 8.25–8.23 (dd, *J* = 8.2, 1.5 Hz, 1H, CH Ar), 8.11–8.07 (m, 3H, CH Ar), 7.74–7.72 (ddd, *J* = 8.4, 6.9, 1.5 Hz, 1H, CH Ar), 7.53–7.49 (m, 3H, CH Ar), 7.13 (s, 1H, CH), 4.30–4.28 (t, *J* = 6.4 Hz, 2H, CH_2_), 2.03–1.98 (m, 2H, CH_2_), 1.63–1.58 (p, *J* = 7.4 Hz, 2H, CH_2_), 1.48–1.44 (dq, *J* = 9.2, 6.7 Hz, 2H, CH_2_), 1.41–1.37 (m, 2H, CH_2_), 1.36–1.32 (qd, *J* = 6.6, 2.7 Hz, 4H, CH_2_), 0.94–0.92 (t, *J* = 6.8 Hz, 3H, CH_3_). ^13^C NMR (151 MHz, CDCl_3_) δ 162.46 (CO), 157.49, 149.16, 138.83, 135.38, 130.08, 129.12, 128.90, 128.86, 125.47, 121.80, 120.57, 98.17, 68.54, 31.83, 29.35, 29.25, 28.97, 26.16, 22.68, 14.13. HRMS Exact mass calcd. for C_23_H_27_ONCl [M + H]^+^: 368.17757; found: 368.17725.

*2-(4-Bromophenyl)-1-octylquinolin-4(1H)-one* (**4d**). Yellow solid; Yield: 70%; Mp: 84.2–86.4 °C; IR (KBr plate) ν_max_ 2921, 2852, 1652, 1592, 1361, 824, 668. ^1^H NMR (600 MHz, CDCl_3_) δ 8.25–8.23(dd, *J* = 8.2, 1.5 Hz, 1H, CH Ar), 8.11–8.10 (d, *J* = 8.4 Hz, 1H, CH Ar), 8.03–8.01 (m, 2H, CH Ar), 7.75–7.72 (ddd, *J* = 8.4, 6.8, 1.5 Hz, 1H, CH Ar), 7.67–7.65 (m, 2H, CH Ar), 7.53–7.50 (ddd, *J* = 8.2, 6.8, 1.2 Hz, 1H, CH Ar), 7.13 (s, 1H, CH), 4.30–4.28 (t, *J* = 6.4 Hz, 2H, CH_2_), 2.03–1.98 (m, 2H, CH_2_), 1.63–1.58 (p, *J* = 7.4 Hz, 2H, CH_2_), 1.48–1.43 (p, *J* = 6.6 Hz, 2H, CH_2_), 1.41–1.37 (m, 2H, CH_2_), 1.36–1.32 (qd, *J* = 6.6, 2.7 Hz, 4H, CH_2_), 0.94–0.92 (t, *J* = 6.8 Hz, 3H, CH_3_). ^13^C NMR (151 MHz, CDCl_3_) δ 162.50 (CO), 157.52, 149.14, 139.25, 131.86, 130.11, 129.15, 129.11, 125.50, 123.75, 121.81, 120.59, 98.12, 68.56, 31.83, 29.34, 29.25, 28.97, 26.16, 22.68, 14.13. HRMS Exact mass calcd. for C_23_H_27_ONBr [M + H]^+^: 412.12705; found: 412.12671.

*1-Octyl-2-p-tolyl-quinolin-4(1H)-one* (**4e**). Yellow solid; Yield: 83%; Mp: 72.0–74.0 °C; IR (KBr plate) ν_max_ 2924, 2853, 1621, 1593, 1356, 816, 612. ^1^H NMR (600 MHz, CDCl_3_) δ 8.24–8.22 (dd, *J* = 8.3, 1.5 Hz, 1H, CH Ar), 8.12–8.11 (d, *J* = 8.4 Hz, 1H, CH Ar), 8.04–8.02 (d, *J* = 8.1 Hz, 2H, CH Ar), 7.73–7.70 (ddd, *J* = 8.4, 6.8, 1.5 Hz, 1H, CH Ar), 7.51–7.48 (ddd, *J* = 8.1, 6.9, 1.2 Hz, 1H, CH Ar), 7.35–7.33 (m, 2H, CH Ar), 7.17 (s, 1H, CH), 4.30–4.28 (t, *J* = 6.4 Hz, 2H, CH_2_), 2.46 (s, 3H, CH_3_), 2.02–1.98 (m, 2H, CH_2_), 1.63–1.58 (p, *J* = 7.4 Hz, 2H, CH_2_), 1.48–1.43 (m, 2H, CH_2_), 1.41–1.37 (m, 2H, CH_2_), 1.36–1.33 (tt, *J* = 6.9, 3.1 Hz, 4H, CH_2_), 0.94–0.92 (t, *J* = 6.8 Hz, 3H, CH_3_). ^13^C NMR (151 MHz, CDCl_3_) δ 162.24 (CO), 158.84, 149.19, 139.28, 137.61, 129.86, 129.47, 129.04, 127.47, 125.10, 121.74, 120.50, 98.42, 68.43, 31.83, 29.35, 29.26, 28.99, 26.17, 22.68, 21.35, 14.13. HRMS Exact mass calcd. for C_24_H_29_ON [M + H]^+^: 348.23219; found: 348.23172.

*2-(4-Ethylphenyl)-1-octylquinolin-4(1H)-one* (**4f**). White solid; Yield: 81%; Mp: 59.4–62.6 °C; IR (KBr plate) ν_max_ 2924, 2850, 1613, 1593, 1360, 824, 616. ^1^H NMR (600 MHz, CDCl_3_) δ 8.24–8.23 (dd, *J* = 8.3, 1.3 Hz, 1H, CH Ar), 8.12–8.11(d, *J* = 8.4 Hz, 1H, CH Ar), 8.06–8.04 (m, 2H, CH Ar), 7.73–7.70 (ddd, *J* = 8.4, 6.8, 1.5 Hz, 1H, CH Ar), 7.51–7.48 (ddd, *J* = 8.1, 6.8, 1.2 Hz, 1H, CH Ar), 7.38–7.37 (d, *J* = 8.2 Hz, 2H, CH Ar), 7.17 (s, 1H, CH), 4.30–4.28 (t, *J* = 6.4 Hz, 2H, CH_2_), 2.78–2.74 (q, *J* = 7.6 Hz, 2H, CH_2_), 2.02–1.98 (m, 2H, CH_2_), 1.63–1.58 (p, *J* = 7.4 Hz, 2H, CH_2_), 1.48–1.42 (m, 2H, CH_2_), 1.41–1.37 (m, 2H, CH_2_), 1.37–1.33(td, *J* = 8.0, 7.0, 3.1 Hz, 4H, CH_2_), 1.33–1.30 (t, *J* = 7.6 Hz, 3H, CH_3_), 0.94–0.92 (t, *J* = 6.9 Hz, 3H, CH_3_). ^13^C NMR (151 MHz, CDCl_3_) δ 162.20 (CO), 158.93, 149.24, 145.62, 137.95, 129.84, 129.09, 128.29, 127.58, 125.09, 121.74, 120.49, 98.49, 68.42, 31.84, 29.36, 29.26, 28.99, 28.74, 26.17, 22.69, 15.62, 14.13. HRMS Exact mass calcd. for C_25_H_32_ON [M + H]^+^: 362.24784; found: 362.24728.

*2-Naphthalen-1-yl-1-octylquinolin-4(1H)-one* (**4g**). Colorless oil; Yield: 62%; IR (KBr plate) ν_max_ 2940, 2851, 1619, 1591, 1361, 836, 648. ^1^H NMR (600 MHz, CDCl_3_) δ 8.33–8.32(dd, *J* = 8.3, 1.5 Hz, 1H, CH Ar), 8.17–8.13 (dd, *J* = 19.8, 8.4 Hz, 2H, CH Ar), 7.97–7.95 (t, *J* = 8.1 Hz, 2H, CH Ar), 7.79–7.76 (ddd, *J* = 8.4, 6.8, 1.5 Hz, 1H, CH Ar), 7.73–7.71 (dd, *J* = 7.0, 1.3 Hz, 1H, CH Ar), 7.62–7.57 (m, 2H, CH Ar), 7.55–7.52 (ddd, *J* = 8.1, 6.7, 1.3 Hz, 1H, CH Ar), 7.50–7.47 (ddd, *J* = 8.3, 6.7, 1.4 Hz, 1H, CH Ar), 7.04 (s, 1H, CH), 4.25–4.22 (t, *J* = 6.5 Hz, 2H, CH_2_), 2.00–1.96 (m, 2H, CH_2_), 1.61–1.56 (dt, *J* = 15.4, 7.4 Hz, 2H, CH_2_), 1.45–1.40 (m, 2H, CH_2_), 1.39–1.31 (m, 6H, CH_2_), 0.92 (t, *J* = 6.9 Hz, 3H, CH_3_). ^13^C NMR (151 MHz, CDCl_3_) δ 161.75 (CO), 160.61, 149.02, 139.50, 133.96, 131.31, 130.02, 129.17, 128.93, 128.35, 127.24, 126.49, 125.93, 125.85, 125.55, 125.35, 121.84, 120.39, 102.84, 68.64, 31.82, 29.34, 29.24, 28.95, 26.13, 22.68, 14.13. HRMS Exact mass calcd. for C_27_H_30_ON [M + H]^+^: 384.23329; found: 384.23187.

*2-Thiophen-2-yl-1-octylquinolin-4(1H)-one* (**4h**). White solid; Yield: 69%; Mp: 40.2–43.4 °C; IR (KBr plate) ν_max_ 2925, 2854, 1616, 1591, 1361, 826, 668. ^1^H NMR (600 MHz, CDCl_3_) δ 8.19–8.17 (dd, *J* = 8.3, 1.5 Hz, 1H, CH Ar), 8.05–8.04 (d, *J* = 8.4 Hz, 1H, CH Ar), 7.72–7.68 (m, 2H, CH Ar), 7.48–7.45 (m, 2H, CH Ar), 7.17–7.16 (dd, *J* = 5.0, 3.7 Hz, 1H, CH Ar), 7.11 (s, 1H, CH), 4.26–4.24 (t, *J* = 6.4 Hz, 2H, CH_2_), 2.01–1.96 (m, 2H, CH_2_), 1.62–1.57 (p, *J* = 7.9, 7.4 Hz, 2H, CH_2_), 1.47–1.45 (m, 2H, CH_2_), 1.41–1.38 (m, 2H, CH_2_), 1.37–1.33 (td, *J* = 10.9, 10.0, 4.5 Hz, 4H, CH_2_), 0.95–0.93 (t, *J* = 6.7 Hz, 3H, CH_3_). ^13^C NMR (151 MHz, CDCl_3_) δ 162.09 (CO), 153.40, 149.08, 145.93, 130.01, 128.77, 128.27, 127.88, 125.43, 125.11, 121.79, 120.75, 96.97, 68.50, 31.85, 29.38, 29.27, 28.97, 26.16, 22.71, 14.16. HRMS Exact mass calcd. for C_21_H_26_ONS [M + H]^+^: 340.17296; found: 340.17264.

*2-Cyclohexyl-1-octylquinolin-4(1H)-one* (**4i**). Colorless oil; Yield: 41%; IR (KBr plate) ν_max_ 2910, 2821, 1639, 1599, 1368, 831, 628. ^1^H NMR (600 MHz, CDCl_3_) δ 8.18–8.17 (d, *J* = 8.1 Hz, 1H, CH Ar), 8.00–7.99 (d, *J* = 8.4 Hz, 1H, CH Ar), 7.68–7.65 (t, *J* = 7.6 Hz, 1H, CH Ar), 7.46–7.43 (t, *J* = 7.7 Hz, 1H, CH Ar), 6.65 (s, 1H, CH), 4.22–4.20 (t, *J* = 6.4 Hz, 2H, CH_2_), 2.91–2.86 (tt, *J* = 12.1, 3.4 Hz, 1H, CH), 2.06–2.04 (d, *J* = 11.2 Hz, 2H, CH_2_), 1.98–1.90 (m, 5H, CH_2_), 1.82–1.80 (d, *J* = 12.8 Hz, 1H, CH_2_), 1.67–1.56 (m, 4H, CH_2_), 1.53–1.41 (m, 4H, CH_2_), 1.40–1.33 (m, 6H, CH_2_), 0.92 (t, *J* = 6.8 Hz, 3H, CH_3_). ^13^C NMR (151 MHz, CDCl_3_) δ 168.19 (CO), 161.98, 148.66, 129.56, 128.31, 124.63, 121.69, 120.43, 98.50, 68.24, 48.20, 32.96, 31.83, 29.35, 29.24, 28.98, 26.57, 26.16, 26.13, 22.68, 14.12. HRMS Exact mass calcd. for C_23_H_34_ON [M + H]^+^: 340.26349; found: 340.26306.

*2-(4-Chlorophenyl)-7-fluoro-1-octylquinolin-4(1H)-one* (**4j**). Yellow solid; Yield: 65%; Mp: 73.3–75.8 °C; IR (KBr plate) ν_max_ 2919, 2851, 1628, 1594, 1354, 815, 667. ^1^H NMR (600 MHz, CDCl_3_) δ 8.24–8.21 (dd, *J* = 9.1, 6.2 Hz, 1H, CH Ar), 8.08–8.06 (m, 2H, CH Ar), 7.73–7.70 (dd, *J* = 10.4, 2.5 Hz, 1H, CH Ar), 7.52–7.49 (m, 2H, CH Ar), 7.28–7.26 (m, 1H, CH Ar), 7.10 (s, 1H, CH), 4.30–4.28 (t, *J* = 6.4 Hz, 2H, CH_2_), 2.02–1.97 (m, 2H, CH_2_), 1.62–1.57 (p, *J* = 7.4 Hz, 2H, CH_2_), 1.47–1.42 (m, 2H, CH_2_), 1.41–1.37 (m, 2H, CH_2_), 1.36–1.32 (qd, *J* = 6.7, 2.8 Hz, 4H, CH_2_), 0.94–0.91 (t, *J* = 6.9 Hz, 3H, CH_3_). ^13^C NMR (151 MHz, CDCl_3_) δ 163.79 (q, ^1^*J*_CF_ = 249.15), 162.55 (CO), 158.75, 150.46 (q, ^3^*J*_CF_ = 12.08), 138.43, 135.68, 128.95, 128.86, 124.17 (q, ^3^*J*_CF_ = 10.57), 117.49, 115.51 (q, ^2^*J*_CF_ = 25.67), 112.79 (q, ^2^*J*_CF_ = 19.63), 97.73, 68.67, 31.82, 29.33, 29.24, 28.93, 26.13, 22.67, 14.12. HRMS Exact mass calcd. for C_23_H_26_ONClF [M + H]^+^: 386.16815; found: 386.16776.

*6-Chloro-2-(4-chlorophenyl)-1-octylquinolin-4(1H)-one* (**4k**). White solid; Yield: 67%; Mp: 92.6–99.2 °C; IR (KBr plate) ν_max_ 2924, 2851, 1616, 1592, 1353, 828, 667. ^1^H NMR (600 MHz, CDCl_3_) δ 8.18 (d, *J* = 2.4 Hz, 1H, CH Ar), 8.08–8.05 (m, 2H, CH Ar), 8.03–8.01 (d, *J* = 8.9 Hz, 1H, CH Ar), 7.66–7.64 (dd, *J* = 9.0, 2.4 Hz, 1H, CH Ar), 7.51–7.49 (m, 2H, CH Ar), 7.14 (s, 1H, CH), 4.29–4.27 (t, *J* = 6.5 Hz, 2H, CH_2_), 2.02–1.98 (m, 2H, CH_2_), 1.62–1.57 (p, *J* = 7.4 Hz, 2H, CH_2_), 1.48–1.43 (dq, *J* = 9.3, 6.7 Hz, 2H, CH_2_), 1.42–1.38 (m, 2H, CH_2_), 1.36–1.32 (qd, *J* = 6.5, 2.7 Hz, 4H, CH_2_), 0.94–0.92 (t, *J* = 6.8 Hz, 3H, CH_3_). ^13^C NMR (151 MHz, CDCl_3_) δ 161.63 (CO), 157.68, 147.55, 138.38, 135.63, 131.27, 130.90, 130.78, 128.96, 128.78, 121.26, 121.03, 98.72, 68.80, 31.83, 29.33, 29.22, 28.90, 26.13, 22.68, 14.13. HRMS Exact mass calcd. for C_23_H_26_ONCl_2_ [M + H]^+^: 402.13860; found: 402.13818.

*6-Bromo-2-(4-chlorophenyl)-1-octylquinolin-4(1H)-one* (**4l**). White solid; Yield: 75%; Mp: 98.9–100.9 °C; IR (KBr plate) ν_max_ 2924, 2850, 1652, 1557, 1361, 827, 668. ^1^H NMR (600 MHz, CDCl_3_) δ 8.35 (d, *J* = 2.3 Hz, 1H, CH Ar), 8.07–8.05 (m, 2H, CH Ar), 7.95–7.94 (d, *J* = 8.9 Hz, 1H, CH Ar), 7.79–7.77 (dd, *J* = 8.9, 2.3 Hz, 1H, CH Ar), 7.51–7.49 (m, 2H, CH Ar), 7.13 (s, 1H, CH), 4.28–4.26 (t, *J* = 6.5 Hz, 2H, CH_2_), 2.02–1.97 (m, 2H, CH_2_), 1.62–1.57 (p, *J* = 7.2 Hz, 2H, CH_2_), 1.48–1.43 (m, 2H, CH_2_), 1.42–1.37 (m, 2H, CH_2_), 1.37–1.32 (qd, *J* = 6.5, 2.7 Hz, 4H, CH_2_), 0.94–0.92 (t, *J* = 6.8 Hz, 3H, CH_3_). ^13^C NMR (151 MHz, CDCl_3_) δ 161.52 (CO), 157.80, 147.75, 138.36, 135.67, 133.47, 130.90, 128.97, 128.79, 124.34, 121.74, 119.33, 98.72, 68.83, 31.83, 29.33, 29.22, 28.89, 26.12, 22.68, 14.14. HRMS Exact mass calcd. for C_23_H_26_ONBrCl [M + H]^+^: 446.08808; found: 446.08771.

*2-(4-Chlorophenyl)-7-methyl-1-octylquinolin-4(1H)-one* (**4m**). Yellow solid; Yield: 76%; Mp: 90.9–93.9 °C; IR (KBr plate) ν_max_ 2965, 2832, 1612, 1593, 1356, 833, 622. ^1^H NMR (600 MHz, CDCl_3_) δ 8.12–8.11 (d, *J* = 8.4 Hz, 1H, CH Ar), 8.08–8.05 (m, 2H, CH Ar), 7.89 (s, 1H, CH Ar), 7.51–7.48 (m, 2H, CH Ar), 7.35–7.33 (dd, *J* = 8.4, 1.7 Hz, 1H, CH Ar), 7.07 (s, 1H, CH), 4.28–4.26 (t, *J* = 6.4 Hz, 2H, CH_2_), 2.58 (s, 3H, CH_3_), 2.01–1.97 (m, 2H, CH_2_), 1.62–1.57 (p, *J* = 7.4 Hz, 2H, CH_2_), 1.47–1.42 (m, 2H, CH_2_), 1.41–1.37 (m, 2H, CH_2_), 1.36–1.32 (m, 4H, CH_2_), 0.94–0.92 (t, *J* = 6.9 Hz, 3H, CH_3_). ^13^C NMR (151 MHz, CDCl_3_) δ 162.44 (CO), 157.47, 149.45, 140.33, 138.95, 135.25, 128.85, 128.81, 128.24, 127.62, 121.50, 118.45, 97.60, 68.44, 31.83, 29.35, 29.25, 28.98, 26.15, 22.68, 21.82, 14.13. HRMS Exact mass calcd. for C_24_H_29_ONCl [M + H]^+^: 382.19322; found: 382.19247.

*7-Fluoro-2-(4-ethylphenyl)-1-octylquinolin-4(1H)-one* (**4n**). Yellow solid; Yield: 66%; Mp: 57.4–60.9 °C; IR (KBr plate) ν_max_ 2967, 2842, 1613, 1598, 1360, 829, 685. ^1^H NMR (600 MHz, CDCl_3_) δ 8.21 (dd, *J* = 9.1, 6.2 Hz, 1H, CH Ar), 8.04 (d, *J* = 8.1 Hz, 2H, CH Ar), 7.73 (dd, *J* = 10.5, 2.6 Hz, 1H, CH Ar), 7.37 (d, *J* = 8.0 Hz, 2H, CH Ar), 7.25 (td, *J* = 8.6, 8.2, 2.5 Hz, 1H, CH Ar), 7.13 (s, 1H, CH), 4.28 (t, *J* = 6.4 Hz, 2H, CH_2_), 2.76 (q, *J* = 7.6 Hz, 2H, CH_2_), 2.02–1.96 (m, 2H, CH_2_), 1.60 (p, *J* = 7.3 Hz, 2H, CH_2_), 1.48–1.42 (m, 2H, CH_2_), 1.39 (dd, *J* = 15.0, 7.2 Hz, 2H, CH_2_), 1.37–1.33 (m, 4H, CH_2_), 1.31 (d, *J* = 7.5 Hz, 3H, CH_3_), 0.93 (t, *J* = 6.8 Hz, 3H, CH_3_). ^13^C NMR (151 MHz, CDCl_3_) δ 163.69 (q, ^1^*J*_CF_ = 249.15), 162.27 (CO), 160.20, 150.55 (q, ^3^*J*_CF_ = 12.08), 145.94, 137.57, 128.33, 127.58, 124.07 (q, ^3^*J*_CF_ = 10.57), 117.41, 115.06 (q, ^2^*J*_CF_ = 25.67), 112.76 (q, ^2^*J*_CF_ = 19.63), 98.01, 68.54, 31.83, 29.34, 29.25, 28.95, 28.75, 26.15, 22.68, 15.59, 14.12. HRMS Exact mass calcd. for C_25_H_31_ONF [M + H]^+^: 380.23842; found: 380.23761.

*6-Chloro-2-(4-ethylphenyl)-1-octylquinolin-4(1H)-one* (**4o**). Yellow solid; Yield: 68%; Mp: 66.6–70.6 °C; IR (KBr plate) ν_max_ 2927, 2856, 1619, 1590, 1365, 824, 635. ^1^H NMR (600 MHz, CDCl_3_) δ 8.18–8.17 (d, *J* = 2.4 Hz, 1H, CH Ar), 8.04–8.02 (dd, *J* = 8.6, 2.9 Hz, 3H, CH Ar), 7.64–7.63 (dd, *J* = 8.9, 2.4 Hz, 1H, CH Ar), 7.37–7.36 (d, *J* = 8.3 Hz, 2H, CH Ar), 7.18 (s, 1H, CH), 4.29–4.27 (t, *J* = 6.5 Hz, 2H, CH_2_), 2.77–2.74 (q, *J* = 7.6 Hz, 2H, CH_2_), 2.02–1.97 (m, 2H, CH_2_), 1.62–1.57 (p, *J* = 7.2 Hz, 2H, CH_2_), 1.48–1.43 (m, 2H, CH_2_), 1.42–1.37 (m, 2H, CH_2_), 1.37–1.33 (ddt, *J* = 10.0, 6.8, 3.1 Hz, 4H, CH_2_), 1.33–1.30 (t, *J* = 7.6 Hz, 3H, CH_3_), 0.94–0.92 (t, *J* = 6.8 Hz, 3H, CH_3_). ^13^C NMR (151 MHz, CDCl_3_) δ 161.38 (CO), 159.14, 147.62, 145.91, 137.48, 130.83, 130.73, 130.65, 128.35, 127.52, 121.18, 120.97, 99.04, 68.67, 31.84, 29.34, 29.23, 28.91, 28.74, 26.14, 22.68, 15.58, 14.13. HRMS Exact mass calcd. for C_25_H_31_ONCl [M + H]^+^: 396.20887; found: 396.20831.

*6-Bromo-2-(4-ethylphenyl)-1-octylquinolin-4(1H)-one* (**4p**). Yellow solid; Yield: 65%; Mp: 57.2–62.3 °C; IR (KBr plate) ν_max_ 2970, 2864, 1653, 1596, 1366, 843, 682. ^1^H NMR (600 MHz, CDCl_3_) δ 8.35 (d, *J* = 2.3 Hz, 1H, CH Ar), 8.04–8.03 (d, *J* = 8.3 Hz, 2H, CH Ar), 7.97–7.96 (d, *J* = 8.9 Hz, 1H, CH Ar), 7.77–7.76 (dd, *J* = 8.9, 2.3 Hz, 1H, CH Ar), 7.37–7.36 (d, *J* = 8.2 Hz, 2H, CH Ar), 7.18 (s, 1H, CH), 4.29–4.27 (t, *J* = 6.5 Hz, 2H, CH_2_), 2.77–2.73 (q, *J* = 7.6 Hz, 2H, CH_2_), 2.02–1.97 (m, 2H, CH_2_), 1.62–1.57 (p, *J* = 7.3 Hz, 2H, CH_2_), 1.48–1.43 (m, 2H, CH_2_), 1.42–1.37 (m, 2H, CH_2_), 1.37–1.33 (ddt, *J* = 9.9, 6.7, 3.2 Hz, 4H, CH_2_), 1.32–1.30 (t, *J* = 7.6 Hz, 3H, CH_3_), 0.94–0.92 (t, *J* = 6.8 Hz, 3H, CH_3_). ^13^C NMR (151 MHz, CDCl_3_) δ 161.27 (CO), 159.26, 147.84, 145.95, 137.48, 133.21, 130.89, 128.36, 127.52, 124.27, 121.68, 118.87, 99.03, 68.70, 31.84, 29.34, 29.23, 28.90, 28.75, 26.13, 22.69, 15.58, 14.13. HRMS Exact mass calcd. for C_25_H_31_ONBr [M + H]^+^: 440.15835; found: 440.15790.

*7-Methyl-2-(4-ethylphenyl)-1-octylquinolin-4(1H)-one* (**4q**). Yellow solid; Yield: 80%; Mp: 65.9–69.8 °C; IR (KBr plate) ν_max_ 2960, 2854, 1643, 1590, 1359, 853, 652. ^1^H NMR (600 MHz, CDCl_3_) δ 8.12–8.10 (d, *J* = 8.3 Hz, 1H, CH Ar), 8.04–8.02 (m, 2H), CH Ar, 7.90 (s, 1H, CH Ar), 7.37–7.36 (d, *J* = 8.3 Hz, 2H, CH Ar), 7.33–7.31 (dd, *J* = 8.4, 1.7 Hz, 1H, CH Ar), 7.11 (s, 1H, CH), 4.29–4.26 (t, *J* = 6.4 Hz, 2H, CH_2_), 2.77–2.73 (q, *J* = 7.6 Hz, 2H, CH_2_), 2.57 (s, 3H, CH_3_), 2.01–1.96 (m, 2H, CH_2_), 1.62–1.57 (p, *J* = 7.3 Hz, 2H, CH_2_), 1.47–1.42 (dt, *J* = 15.0, 6.7 Hz, 2H, CH_2_), 1.41–1.37 (m, 2H, CH_2_), 1.36–1.32 (m, 4H, CH_2_), 1.31–1.28 (d, *J* = 7.6 Hz, 3H, CH_3_), δ 1.33–1.30 (t, *J* = 7.6 Hz, 3H, CH_3_),0.94–0.92 (t, *J* = 6.9 Hz, 3H). ^13^C NMR (151 MHz, CDCl_3_) δ 162.20 (CO), 158.91, 149.50, 145.51, 140.03, 138.04, 128.24, 128.22, 127.54, 127.23, 121.45, 118.36, 97.92, 68.33, 31.83, 29.36, 29.26, 29.00, 28.73, 26.16, 22.68, 21.82, 15.62, 14.13. HRMS Exact mass calcd. for C_26_H_34_ON [M + H]^+^: 376.26349; found: 376.26254.

*1-Methyl-2-phenylquinolin-4(1H)-one* (**4r**) [66]. Yellow solid; Yield:73 %; Mp: 62.1–64.1 °C; IR (KBr plate) ν_max_ 2987, 2863, 1652, 1591, 1365, 844, 688. ^1^H NMR (600 MHz, CDCl_3_) δ 8.23–8.22 (dd, *J* = 8.3, 1.5 Hz, 1H, CH Ar), 8.15–8.13 (m, 3H, CH Ar), 7.75–7.72 (ddd, *J* = 8.4, 6.8, 1.5 Hz, 1H, CH Ar), 7.56–7.48 (m, 4H, CH Ar), 7.21 (s, 1H, CH), 4.15 (s, 3H, CH_3_). ^13^C NMR (151 MHz, CDCl_3_) δ 162.88 (CO), 158.89, 149.16, 140.39, 130.01, 129.28, 129.18, 128.78, 127.61, 125.43, 121.64, 120.40, 98.04, 55.69. HRMS Exact mass calcd. for C_16_H_14_ON [M + H]^+^: 236.10699; found: 236.10645.

*2-(4-Chlorophenyl)-1-methylquinolin-4(1H)-one* (**4s**) [67]. Yellow solid; Yield: 63%; Mp: 109.1–110.9 °C; IR (KBr plate) ν_max_ 2917, 2853, 1652, 1593, 1360, 824, 668. ^1^H NMR (600 MHz, CDCl_3_) δ 8.22–8.21 (d, *J* = 8.2 Hz, 1H, CH Ar), 8.12–8.08 (t, *J* = 8.6 Hz, 3H, CH Ar), 7.75–7.72 (ddd, *J* = 8.4, 6.8, 1.5 Hz, 1H, CH Ar), 7.53–7.50 (m, 3H, CH Ar), 7.16 (s, 1H, CH), 4.15 (s, 3H, CH_3_). ^13^C NMR (151 MHz, CDCl_3_) δ 163.03 (CO), 157.49, 149.11, 138.74, 135.45, 130.17, 129.14, 128.93, 128.86, 125.62, 121.69, 120.43, 97.59, 55.72. HRMS Exact mass calcd. for C_16_H_13_ONCl [M + H]^+^: 270.06802; found: 270.06750.

*2-(4-Ethylphenyl)-1-methylquinolin-4(1H)-one* (**4t**). Colorless oil; Yield: 74%; IR (KBr plate) ν_max_ 2963, 2860, 1683, 1592, 1356, 827, 612. ^1^H NMR (600 MHz, CDCl_3_) δ 8.22–8.20 (dd, *J* = 8.3, 0.9 Hz, 1H, CH Ar), 8.13–8.12 (d, *J* = 8.2 Hz, 1H, CH Ar), 8.07–8.05 (m, 2H, CH Ar), 7.74–7.71 (ddd, *J* = 8.4, 6.8, 1.5 Hz, 1H, CH Ar), 7.51–7.48 (ddd, *J* = 8.1, 6.8, 1.2 Hz, 1H, CH Ar), 7.39–7.37 (d, *J* = 8.5 Hz, 2H, CH Ar), 7.19 (s, 1H, CH), 4.14 (s, 3H, CH_3_), 2.78–2.74 (q, *J* = 7.6 Hz, 2H), 1.33–1.30 (t, *J* = 7.6 Hz, 3H). ^13^C NMR (151 MHz, CDCl_3_) δ 162.77 (CO), 158.92, 149.19, 145.70, 137.85, 129.92, 129.12, 128.32, 127.58, 125.23, 121.61, 120.34, 97.90, 55.64, 28.74, 15.61. HRMS Exact mass calcd. for C_18_H_18_ON [M + H]^+^: 264.13829; found: 264.13763.

*7-Fluoro-1-methyl-2-phenylquinolin-4(1H)-one* (**4u**). White solid; Yield: 50 %; Mp: 61.3–63.4 °C; IR (KBr plate) ν_max_ 2970, 2833, 1654, 1590, 1369, 840, 648. ^1^H NMR (600 MHz, CDCl_3_) δ 8.21 (dd, *J* = 9.1, 6.2 Hz, 1H, CH Ar), 8.14–8.11 (m, 2H, CH Ar), 7.76–7.74 (dd, *J* = 10.4, 2.6 Hz, 1H, CH Ar), 7.57–7.54 (m, 2H, CH Ar), 7.51–7.49 (m, 1H, CH Ar), 7.28–7.26 (m, 1H, CH Ar), 7.17 (s, 1H, CH), 4.15 (s, 3H, CH_3_). ^13^C NMR (151 MHz, CDCl_3_) δ 163.77 (q, ^1^*J*_CF_ = 249.15), 162.92 (CO), 160.20, 150.45, 140.03, 129.55, 128.82, 127.61, 124.01 (q, ^3^*J*_CF_ = 9.06), 117.32, 115.45 (q, ^2^*J*_CF_ = 24.16), 112.86 (q, ^2^*J*_CF_ = 21.14), 97.60, 55.75. HRMS Exact mass calcd. for C_16_H_13_ONF [M + H]^+^: 254.09757; found: 254.09703.

*7-Fluoro-2-(4-ethylphenyl)-1-methylquinolin-4(1H)-one* (**4v**). Yellow solid; Yield: 53%; Mp: 55.2–59.1 °C; IR (KBr plate) ν_max_ 2960, 2843, 1634, 1590, 1359, 850, 628. ^1^H NMR (600 MHz, CDCl_3_) δ 8.21–8.18 (dd, *J* = 9.1, 6.2 Hz, 1H, CH Ar), 8.05–8.04 (d, 2H, CH Ar), 7.74–7.72 (dd, *J* = 10.5, 2.5 Hz, 1H, CH Ar), 7.38–7.37 (d, *J* = 8.3 Hz, 2H, CH Ar), 7.27–7.24 (ddd, *J* = 9.1, 8.2, 2.5 Hz, 1H, CH Ar), 7.15 (s, 1H, CH), 4.13 (s, 3H, CH_3_), 2.78–2.74 (q, *J* = 7.6 Hz, 2H), 1.33–1.30 (t, *J* = 7.6 Hz, 3H). ^13^C NMR (151 MHz, CDCl_3_) δ 163.71 (q, ^1^*J*_CF_ = 249.15), 162.83 (CO), 160.21, 150.54 (q, ^3^*J*_CF_ = 12.08), 146.01, 137.50, 128.36, 127.57, 123.96 (q, ^3^*J*_CF_ = 10.57), 117.26, 115.21 (q, ^2^*J*_CF_ = 25.67), 112.8 1(q, ^2^*J*_CF_ = 19.63), 97.41, 55.70, 28.75, 15.59. HRMS Exact mass calcd. for C_18_H_17_ONF [M + H]^+^: 282.12887; found: 282.12817.

*2-(4-Ethylphenyl)-1,7-dimethylquinolin-4(1H)-one* (**4w**). White solid; Yield: 78%; Mp: 86.1–88.3 °C; IR (KBr plate) ν_max_ 2925, 2863, 1625, 1596, 1353, 816, 640. ^1^H NMR (600 MHz, CDCl_3_) δ 8.10–8.08 (d, *J* = 8.4 Hz, 1H, CH Ar), 8.05–8.04 (d, *J* = 8.2 Hz, 2H, CH Ar), 7.91 (s, 1H, CH Ar), 7.38–7.36 (d, *J* = 8.1 Hz, 2H, CH Ar), 7.34–7.32 (dd, *J* = 8.4, 1.7 Hz, 1H, CH Ar), 7.13 (s, 1H, CH), 4.13 (s, 3H, CH_3_), 2.77–2.74 (q, *J* = 7.6 Hz, 2H, CH_2_), 2.58 (s, 3H, CH_3_), 1.33–1.30 (t, *J* = 7.6 Hz, 3H, CH_3_). ^13^C NMR (151 MHz, CDCl_3_) δ 162.79 (CO), 158.91, 149.43, 145.60, 140.17, 137.92, 128.27, 128.21, 127.54, 127.41, 121.31, 118.21, 97.33, 55.57, 28.73, 21.81, 15.61. HRMS Exact mass calcd. for C_19_H_20_ON [M + H]^+^: 278.15394; found: 278.15338.

*2-(4-Chlorophenyl)-1,7-dimethylquinolin-4(1H)-one* (**4x**). Yellow solid; Yield: 73%; Mp: 102.3–104.1 °C; IR (KBr plate) ν_max_ 2921, 2820, 1626, 1595, 1354, 897, 654. ^1^H NMR (600 MHz, CDCl_3_) δ 8.11–8.07 (m, 3H, CH Ar), 7.90 (s, 1H, CH Ar), 7.52–7.49 (m, 2H, CH Ar), 7.36–7.34 (dd, *J* = 8.4, 1.7 Hz, 1H, CH Ar), 7.10 (s, 1H, CH), 4.14 (s, 3H, CH_3_), 2.58 (s, 3H, CH_3_). ^13^C NMR (151 MHz, CDCl_3_) δ 163.03 (CO), 157.49, 149.37, 140.47, 138.84, 135.34, 128.89, 128.83, 128.23, 127.80, 121.38, 118.30, 97.04, 55.65, 21.82. HRMS Exact mass calcd. for C_17_H_15_ONCl [M + H]^+^: 284.08367; found: 284.08316.

*1-Ethyl-2-phenylquinolin-4(1H)-one* (**4y**) [31]. Yellow solid; Yield: 84%; Mp: 101.4–103.6 °C; IR (KBr plate) ν_max_ 2982, 2843, 1613, 1592, 1352, 825, 694. ^1^H NMR (600 MHz, CDCl_3_) δ 8.26–8.25 (dd, *J* = 8.3, 0.9 Hz, 1H, CH Ar), 8.14–8.12 (m, 3H, CH Ar), 7.74–7.72 (ddd, *J* = 8.4, 6.8, 1.5 Hz, 1H, CH Ar), 7.56–7.47 (m, 4H, CH Ar), 7.19 (s, 1H, CH), 4.40–4.36 (q, *J* = 7.0 Hz, 2H, CH_2_), 1.65–1.62 (t, *J* = 7.0 Hz, 3H, CH_3_). ^13^C NMR (151 MHz, CDCl_3_) δ 162.16 (CO), 158.88, 149.25, 140.49, 129.93, 129.22, 129.18, 128.76, 127.60, 125.28, 121.77, 120.48, 98.59, 64.10, 14.58. HRMS Exact mass calcd. for C_17_H_16_ON [M + H]^+^: 250.112264; found: 250.12204.

*1-Ethyl-2-(4-ethylphenyl)quinolin-4(1H)-one* (**4z**). Yellow solid; Yield: 82%; Mp: 99.8–101.2 °C; IR (KBr plate) ν_max_ 2958, 2865, 1652, 1592, 1353, 821, 640. ^1^H NMR (600 MHz, CDCl_3_) δ 8.25–8.23 (dd, *J* = 8.3, 0.9 Hz, 1H, CH Ar), 8.13–8.12 (d, *J* = 8.4 Hz, 1H, CH Ar), 8.06–8.04 (m, 2H, CH Ar), 7.73–7.70 (ddd, *J* = 8.4, 6.8, 1.5 Hz, 1H, CH Ar), 7.51–7.48 (ddd, *J* = 8.2, 6.8, 1.2 Hz, 1H, CH Ar), 7.38–7.37 (d, *J* = 8.3 Hz, 2H, CH Ar), 7.17 (s, 1H, CH), 4.39–4.35 (q, *J* = 7.0 Hz, 2H, CH_2_), 2.78–2.74 (q, *J* = 7.6 Hz, 2H, CH_2_), 1.64–1.62 (t, *J* = 7.0 Hz, 3H, CH_3_), 1.33–1.30 (t, *J* = 7.6 Hz, 3H, CH_3_). ^13^C NMR (151 MHz, CDCl_3_) δ 162.08 (CO), 158.90, 149.22, 145.65, 137.89, 129.87, 129.07, 128.30, 127.58, 125.11, 121.74, 120.42, 98.46, 64.05, 28.74, 15.62, 14.58. HRMS Exact mass calcd. for C_19_H_20_ON [M + H]^+^: 278.15394; found: 278.15341.

*2-(4-Chlorophenyl)-1-ethylquinolin-4(1H)-one* (**4aa**). Yellow solid; Yield: 58%; Mp: 104.3–106.2 °C; IR (KBr plate) ν_max_ 2951, 2834, 1618, 1592, 1359, 834, 641. ^1^H NMR (600 MHz, CDCl_3_) δ 8.25–8.24 (d, *J* = 8.3 Hz, 1H, CH Ar), 8.10–8.07 (t, *J* = 9.0 Hz, 3H, CH Ar), 7.74–7.72 (t, *J* = 7.6 Hz, 1H, CH Ar), 7.50–7.50 (d, *J* = 8.5 Hz, 3H, CH Ar), 7.13 (s, 1H, CH), 4.39–4.35 (q, *J* = 7.0 Hz, 2H, CH_2_), 1.64 (t, *J* = 7.0 Hz, 3H, CH_3_). ^13^C NMR (151 MHz, CDCl_3_) δ 162.31 (CO), 157.47, 149.18, 138.83, 135.37, 130.09, 129.14, 128.91, 128.85, 125.48, 121.81, 120.50, 98.13, 64.17, 14.57. HRMS Exact mass calcd. for C_17_H_15_ONCl [M + H]^+^: 284.08367; found: 284.08331.

*7-Fluoro-1-ethyl-2-phenylquinolin-4(1H)-one* (**4ab**). Yellow solid; Yield: 56%; Mp: 120.3–124.0 °C; IR (KBr plate) ν_max_ 2970, 2865, 1625, 1593, 1355, 819, 693. ^1^H NMR (600 MHz, CDCl_3_) δ 8.25–8.23 (dd, *J* = 9.1, 6.2 Hz, 1H, CH Ar), 8.13–8.09 (m, 2H, CH Ar), 7.75–7.73 (dd, *J* = 10.5, 2.5 Hz, 1H, CH Ar), 7.56–7.53 (t, *J* = 7.3 Hz, 2H, CH Ar), 7.50–7.48 (t, *J* = 7.3 Hz, 1H, CH Ar), 7.28–7.24 (m, 1H, CH Ar), 7.15 (s, 1H, CH), 4.40–4.36 (q, *J* = 7.0 Hz, 2H, CH_2_), 1.64–1.62 (t, *J* = 6.9 Hz, 3H, CH_3_). ^13^C NMR (151 MHz, CDCl_3_) δ 163.72 (q, ^1^*J*_CF_ = 249.15), 162.23 (CO), 160.18, 150.56 (q, ^3^*J*_CF_ = 12.08), 140.14, 129.47, 128.80, 127.60, 124.14(q, ^3^*J*_CF_ = 10.57), 117.40, 115.29 (q, ^2^*J*_CF_ = 25.67), 112.83(q, ^2^*J*_CF_ = 21.14), 98.15, 64.23, 14.55. HRMS Exact mass calcd. for C_17_H_15_ONF [M + H]^+^: 268.11322; found: 268.11258.

*7-Fluoro-2-(4-chlorophenyl)-1-ethylquinolin-4(1H)-one* (**4ac**). Yellow solid; Yield: 73%; Mp: 147.8–154.6 °C; IR (KBr plate) ν_max_ 2987, 2864, 1628, 1594, 1353, 813, 680. ^1^H NMR (600 MHz, CDCl_3_) δ 8.25–8.22 (dd, *J* = 9.1, 6.2 Hz, 1H, CH Ar), 8.07–8.05 (m, 2H, CH Ar), 7.72–7.70 (dd, *J* = 10.4, 2.5 Hz, 1H, CH Ar), 7.51–7.49 (m, 2H, CH Ar), 7.28–7.25 (m, 1H, CH Ar), 7.10 (s, 1H, CH), 4.39–4.35 (q, *J* = 7.0 Hz, 2H, CH_2_), 1.64–1.62 (t, *J* = 7.0 Hz, 3H, CH_3_). ^13^C NMR (151 MHz, CDCl_3_) δ 163.80 (q, ^1^*J*_CF_ = 249.15), 162.41 (CO), 158.77, 150.49 (q, ^3^*J*_CF_ = 10.57), 138.44, 135.68, 128.96, 128.86, 124.20 (q, ^3^*J*_CF_ = 10.57), 117.43, 115.51 (q, ^2^*J*_CF_ = 24.16), 112.78 (q, ^2^*J*_CF_ = 21.14), 97.71, 64.31, 14.53. HRMS Exact mass calcd. for C_17_H_14_ONClF [M + H]^+^: 302.07425 found: 302.07376.

*7-Fluoro-2-(4-ethylphenyl)-1-ethylquinolin-4(1H)-one* (**4ad**). Yellow solid; Yield: 79%; Mp: 87.4–89.3 °C; IR (KBr plate) ν_max_ 2954, 2874, 1628, 1593, 1355, 817, 648. ^1^H NMR (600 MHz, CDCl_3_) δ 8.24–8.21 (dd, *J* = 9.1, 6.2 Hz, 1H, CH Ar), 8.04–8.02 (d, *J* = 8.2 Hz, 2H, CH Ar), 7.74–7.71 (dd, *J* = 10.5, 2.5 Hz, 1H, CH Ar), 7.38–7.36 (d, *J* = 8.0 Hz, 2H, CH Ar), 7.27–7.23 (td, *J* = 8.6, 2.5 Hz, 1H, CH Ar), 7.13 (s, 1H, CH), 4.38–4.35 (q, *J* = 7.0 Hz, 2H, CH_2_), 2.78–2.74 (q, *J* = 7.6 Hz, 2H, CH_2_), 1.63–1.61 (t, *J* = 7.0 Hz, 3H, CH_3_), 1.33–1.30 (t, *J* = 7.6 Hz, 3H). ^13^C NMR (151 MHz, CDCl_3_) δ 163.69 (q, ^1^*J*_CF_ = 249.15), 162.12 (CO), 160.19, 150.58 (q, ^3^*J*_CF_ = 13.59), 145.95, 137.57, 128.34, 127.57, 124.09 (q, ^3^*J*_CF_ = 9.06), 117.34, 115.06 (q, ^2^*J*_CF_ = 24.16), 112.76 (q, ^2^*J*_CF_ = 19.63), 97.98, 64.16, 28.74, 15.59, 14.55. HRMS Exact mass calcd. for C_19_H_19_ONF [M + H]^+^: 296.14352; found: 296.14398.

*7-Methyl-2-(4-ethylphenyl)-1-ethylquinolin-4(1H)-one* (**4ae**). Yellow solid; Yield: 82%; Mp: 89.9–94.1 °C; IR (KBr plate) ν_max_ 2965, 2853, 1624, 1505, 1351, 816, 641. ^1^H NMR (600 MHz, CDCl_3_) δ 8.12 (d, *J* = 8.4 Hz, 1H, CH Ar), 8.03 (d, *J* = 8.2 Hz, 2H, CH Ar), 7.90 (s, 1H, CH Ar), 7.36 (d, *J* = 8.3 Hz, 2H, CH Ar), 7.32 (dd, *J* = 8.4, 1.7 Hz, 1H, CH Ar), 7.11 (s, 1H, CH), 4.36 (q, *J* = 7.0 Hz, 2H, CH_2_), 2.75 (q, *J* = 7.6 Hz, 2H, CH_2_), 2.57 (s, 3H, CH_3_), 1.62 (t, *J* = 7.0 Hz, 3H), 1.31 (t, *J* = 7.6 Hz, 3H). ^13^C NMR (151 MHz, CDCl_3_) δ 162.07 (CO), 158.89, 149.49, 145.53, 140.07, 138.02, 128.25, 128.20, 127.53, 127.25, 121.45, 118.30, 97.90, 63.95, 28.73, 21.81, 15.61, 14.59. HRMS Exact mass calcd. for C_20_H_22_ON [M + H]^+^: 292.16959; found: 292.16885.

*2-Phenyl-1-propylquinolin-4(1H)-one* (**4af**). Yellow solid; Yield: 78%; Mp: 63.8–69.8 °C; IR (KBr plate) ν_max_ 2963, 2865, 1616, 1592, 1361, 832, 692. ^1^H NMR (600 MHz, CDCl_3_) δ 8.27–8.25 (dd, *J* = 8.2, 1.5 Hz, 1H, CH Ar), 8.14–8.11 (m, 3H, CH Ar), 7.74–7.72 (ddd, *J* = 8.4, 6.8, 1.5 Hz, 1H, CH Ar), 7.56–7.47 (m, 4H, CH Ar), 7.19 (s, 1H, CH), 4.29–4.26 (t, *J* = 6.4 Hz, 2H, CH_2_), 2.07–2.01 (h, *J* = 7.4 Hz, 2H, CH_2_), 1.21–1.18 (t, *J* = 7.4 Hz, 3H, CH_3_). ^13^C NMR (151 MHz, CDCl_3_) δ 162.29 (CO), 158.91, 149.22, 140.49, 129.93, 129.21, 129.16, 128.76, 127.61, 125.28, 121.74, 120.54, 98.64, 69.91, 22.42, 10.69. HRMS Exact mass calcd. for C_18_H_18_ON [M + H]^+^: 264.13829; found: 264.13766.

*2-(4-Ethylphenyl)-1-propylquinolin-4(1H)-one* (**4ag**). Yellow solid; Yield: 64%; Mp: 63.6–68.5 °C; IR (KBr plate) ν_max_ 2964, 2836, 1652, 1592, 1360, 825, 668. ^1^H NMR (600 MHz, CDCl_3_) δ 8.25–8.24 (dd, *J* = 8.3, 1.5 Hz, 1H, CH Ar), 8.12–8.11 (d, *J* = 8.4 Hz, 1H, CH Ar), 8.06–8.04 (d, *J* = 8.2 Hz, 2H, CH Ar), 7.73–7.70 (ddd, *J* = 8.4, 6.8, 1.5 Hz, 1H, CH Ar), 7.51–7.48 (ddd, *J* = 8.2, 6.8, 1.2 Hz, 1H, CH Ar), 7.38–7.37 (d, *J* = 8.3 Hz, 2H, CH Ar), 7.18 (s, 1H, CH), 4.28–4.25 (t, *J* = 6.4 Hz, 2H, CH_2_), 2.78–2.74 (q, *J* = 7.6 Hz, 2H, CH_2_), 2.06–2.01 (h, *J* = 7.4 Hz, 2H, CH_2_), 1.33–1.30 (t, *J* = 7.6 Hz, 3H, CH_3_), 1.21–1.18 (t, *J* = 7.4 Hz, 3H, CH_3_). ^13^C NMR (151 MHz, CDCl_3_) δ 162.18 (CO), 158.92, 149.25, 145.63, 137.94, 129.84, 129.11, 128.29, 127.57, 125.09, 121.71, 120.49, 98.48, 69.85, 28.74, 22.43, 15.62, 10.69. HRMS Exact mass calcd. for C_20_H_22_ON [M + H]^+^: 292.16959; found: 292.16882.

*7-Fluoro-2-phenyl-1-propylquinolin-4(1H)-one* (**4ah**). Yellow solid; Yield: 65%; Mp: 89.5–95.0 °C; IR (KBr plate) ν_max_ 2959, 2858, 1625, 1594, 1364, 816, 689. ^1^H NMR (600 MHz, CDCl_3_) δ 8.26–8.23 (dd, *J* = 9.1, 6.3 Hz, 1H, CH Ar), 8.12–8.11 (m, 2H, CH Ar), 7.75–7.73 (dd, *J* = 10.5, 2.5 Hz, 1H, CH Ar), 7.56–7.53 (m, 2H, CH Ar), 7.51–7.48 (m, 1H, CH Ar), 7.28–7.25 (m, 1H, CH Ar), 7.16 (s, 1H, CH), 4.28–4.26 (t, *J* = 6.4 Hz, 2H, CH_2_), 2.06–2.00 (h, *J* = 7.4 Hz, 2H, CH_2_), 1.20–1.18 (t, *J* = 7.4 Hz, 3H, CH_3_). ^13^C NMR (151 MHz, CDCl_3_) δ 163.72 (q, ^1^*J*_CF_ = 249.15), 162.36 (CO), 160.19, 150.55 (q, ^3^*J*_CF_ = 13.59), 140.13, 129.47, 128.80, 127.60, 124.09 (q, ^3^*J*_CF_ = 10.57), 117.46, 115.29 (q, ^2^*J*_CF_ = 24.16), 112.84 (q, ^2^*J*_CF_ = 21.14), 98.18, 70.02, 22.39, 10.66. HRMS Exact mass calcd. for C_18_H_16_ONFNa [M + Na]^+^: 304.11081; found: 304.11011.

*7-Fluoro-2-(4-chlorophenyl)-1-propylquinolin-4(1H)-one* (**4ai**). Yellow solid; Yield: 54%; Mp: 93.2–103.9 °C; IR (KBr plate) ν_max_ 2969, 2854, 1622, 1594, 1361, 826, 681. ^1^H NMR (600 MHz, CDCl_3_) δ 8.25–8.23 (dd, *J* = 9.1, 6.2 Hz, 1H, CH Ar), 8.08–8.06 (d, *J* = 8.5 Hz, 2H, CH Ar), 7.72–7.70 (dd, *J* = 10.3, 2.5 Hz, 1H, CH Ar), 7.51–7.50 (d, *J* = 8.5 Hz, 2H, CH Ar), 7.28–7.26 (dd, *J* = 8.6, 2.0 Hz, 1H, CH Ar), 7.11 (s, 1H, CH), 4.27–4.25 (t, *J* = 6.4 Hz, 2H, CH_2_), 2.06–2.00 (h, *J* = 7.1 Hz, 2H, CH_2_), 1.20–1.18 (t, *J* = 7.4 Hz, 3H, CH_3_). ^13^C NMR (151 MHz, CDCl_3_) δ 163.79 (q, ^1^*J*_CF_ = 249.15), 162.52 (CO), 158.78, 150.50 (q, ^3^*J*_CF_ = 12.08), 138.47, 135.67, 128.95, 128.86, 124.15 (q, ^3^*J*_CF_ = 10.57), 117.49, 115.51 (q, ^2^*J*_CF_ = 24.16), 112.81 (q, ^2^*J*_CF_ = 21.14), 97.75, 70.08, 22.38, 10.65. HRMS Exact mass calcd. for C_18_H_16_ONClF [M + H]^+^: 316.08990; found: 316.08942.

*7-Fluoro-2-(4-ethylphenyl)-1-propylquinolin-4(1H)-one* (**4aj**). Yellow solid; Yield: 50%; Mp: 59.3–67.5 °C; IR (KBr plate) ν_max_ 2964, 2881, 1627, 1593, 1361, 820, 667. ^1^H NMR (600 MHz, CDCl_3_) δ 8.24–8.21 (dd, *J* = 9.1, 6.2 Hz, 1H, CH Ar), 8.04–8.03 (d, *J* = 8.2 Hz, 2H, CH Ar), 7.74–7.72 (dd, *J* = 10.5, 2.5 Hz, 1H, CH Ar), 7.38–7.36 (d, *J* = 8.0 Hz, 2H, CH Ar), 7.27–7.23 (td, *J* = 8.6, 2.6 Hz, 1H, CH Ar), 7.14 (s, 1H, CH), 4.27–4.25 (t, *J* = 6.4 Hz, 2H, CH_2_), 2.78–2.74 (q, *J* = 7.6 Hz, 2H, CH_2_), 2.06–2.00 (h, *J* = 7.1 Hz, 2H, CH_2_), 1.32–1.30 (t, *J* = 7.6 Hz, 3H, CH_3_), 1.20–1.17 (t, *J* = 7.4 Hz, 3H, CH_3_). ^13^C NMR (151 MHz, CDCl_3_) δ 163.39 (q, ^1^*J*_CF_ = 249.15), 162.26 (CO), 160.21, 150.59, 145.95, 137.56, 128.33, 127.58, 124.05 (q, ^3^*J*_CF_ = 10.57), 117.40, 115.08 (q, ^2^*J*_CF_ = 25.67), 112.76 (q, ^2^*J*_CF_ = 19.63), 98.02, 69.96, 28.74, 22.39, 15.59, 10.66. HRMS Exact mass calcd. for C_20_H_20_ONFNa [M + Na]^+^: 332.14211; found: 332.14133.

*7-Methyl-2-phenyl-1-propylquinolin-4(1H)-one* (**4ak**). Yellow solid; Yield: 77%; Mp: 80.1–82.6 °C; IR (KBr plate) ν_max_ 2925, 2843, 1652, 1594, 1361, 814, 694. ^1^H NMR (600 MHz, CDCl_3_) δ 8.14–8.10 (m, 3H, CH Ar), 7.92 (s, 1H, CH Ar), 7.55–7.52 (t, *J* = 7.5 Hz, 2H, CH Ar), 7.49–7.46 (m, 1H, CH Ar), 7.35–7.33 (dd, *J* = 8.4, 1.7 Hz, 1H, CH Ar), 7.13 (s, 1H, CH), 4.27–4.25 (t, *J* = 6.4 Hz, 2H, CH_2_), 2.58 (s, 3H, CH_3_), 2.06–2.00 (h, *J* = 7.4 Hz, 2H, CH_2_), 1.20–1.17 (t, *J* = 7.4 Hz, 3H, CH_3_). ^13^C NMR (151 MHz, CDCl_3_) δ 162.28 (CO), 158.89, 149.49, 140.60, 140.15, 129.11, 128.71, 128.29, 127.57, 127.43, 121.45, 118.42, 98.07, 69.81, 22.43, 21.82, 10.68. HRMS Exact mass calcd. for C_19_H_19_ONNa [M + Na]^+^: 300.13589; found: 300.13525.

*7-Methyl-2-(4-ethylphenyl)-1-propylquinolin-4(1H)-one* (**4al**). Yellow solid; Yield: 80%; Mp: 79.0–83.4 °C; IR (KBr plate) ν_max_ 2968, 2820, 1623, 1593, 1356, 816, 665. ^1^H NMR (600 MHz, CDCl_3_) δ 8.13–8.11 (d, *J* = 8.4 Hz, 1H, CH Ar), 8.04–8.03 (d, *J* = 8.2 Hz, 2H, CH Ar), 7.90 (s, 1H, CH Ar), 7.37–7.36 (d, *J* = 8.2 Hz, 2H, CH Ar), 7.33–7.31 (dd, *J* = 8.4, 1.7 Hz, 1H, CH Ar), 7.12 (s, 1H, CH), 4.26–4.24 (t, *J* = 6.4 Hz, 2H, CH_2_), 2.77–2.73 (q, *J* = 7.6 Hz, 2H, CH_2_), 2.57 (s, 3H, CH_3_), 2.05–1.99 (h, *J* = 7.4 Hz, 2H, CH_2_), 1.32–1,30 (t, *J* = 7.6 Hz, 3H, CH_3_), 1.20–1.17 (t, *J* = 7.4 Hz, 3H, CH_3_). ^13^C NMR (151 MHz, CDCl_3_) δ 162.18 (CO), 158.91, 149.50, 145.51, 140.04, 138.04, 128.24, 128.22, 127.53, 127.24, 121.42, 118.36, 97.93, 69.76, 28.73, 22.43, 21.82, 15.61, 10.68. HRMS Exact mass calcd. for C_21_H_23_ONNa [M + Na]^+^: 328.16719; found: 328.16669.

*2-(4-Eethylphenyl)-1-isopropylquinolin-4(1H)-one* (**4am**). Yellow solid; Yield: 48%; Mp: 86.1–88.3 °C; IR (KBr plate) ν_max_ 2925, 2854, 1652, 1587, 1384, 830, 668. ^1^H NMR (600 MHz, CDCl_3_) δ 8.23–8.22 (dd, *J* = 8.4, 1.5 Hz, 1H, CH Ar), 8.11–8.09 (d, *J* = 8.3 Hz, 1H, CH Ar), 8.03–8.02 (d, *J* = 8.1 Hz, 2H, CH Ar), 7.72–7.69 (ddd, *J* = 8.4, 6.8, 1.5 Hz, 1H, CH Ar), 7.49–7.46 (ddd, *J* = 8.2, 6.8, 1.2 Hz, 1H, CH Ar), 7.38–7.36 (d, *J* = 8.1 Hz, 2H, CH Ar), 7.17 (s, 1H, CH), 5.00–4.94 (hept, *J* = 6.1 Hz, 1H, CH), 2.78–2.74 (q, *J* = 7.6 Hz, 2H, CH_2_), 1.55–1.54 (d, *J* = 6.1 Hz, 6H, CH_3_), 1.33–1.30 (t, *J* = 7.6 Hz, 3H, CH_3_). ^13^C NMR (151 MHz, CDCl_3_) δ 161.05 (CO), 158.89, 149.48, 145.58, 135.41, 129.82, 129.07, 128.29, 127.60, 124.97, 121.96, 99.24, 70.58, 28.73, 21.87, 15.62. HRMS Exact mass calcd. for C_20_H_21_ONNa [M + Na]^+^: 314.15154; found: 314.15082.

*1-Butyl-2-phenylquinolin-4(1H)-one* (**4an**) [68]. Yellow solid; Yield: 71%; Mp: 75.8–77.8 °C; IR (KBr plate) ν_max_ 2951, 2869, 1621, 1593, 1361, 835, 692. ^1^H NMR (600 MHz, CDCl_3_) δ 8.25 (dd, *J* = 8.3, 1.5 Hz, 1H, CH Ar), 8.13 (dd, *J* = 7.0, 1.5 Hz, 3H, CH Ar), 7.73 (ddd, *J* = 8.4, 6.8, 1.5 Hz, 1H, CH Ar), 7.57–7.47 (m, 4H, CH Ar), 7.19 (s, 1H, CH), 4.31 (t, *J* = 6.4 Hz, 2H, CH_2_), 2.02–1.97 (m, 2H, CH_2_), 1.66 (h, *J* = 7.4 Hz, 2H, CH_2_), 1.08 (t, *J* = 7.4 Hz, 3H, CH_3_). ^13^C NMR (151 MHz, CDCl_3_) δ 162.30 (CO), 158.91, 149.23, 140.50, 129.92, 129.21, 129.17, 128.76, 127.61, 125.28, 121.76, 120.55, 98.62, 68.15, 31.05, 19.40, 13.90. HRMS Exact mass calcd. for C_19_H_19_ONNa [M + Na]^+^: 300.13589; found: 300.13525.

*1-Neopentyl-2-phenylquinolin-4(1H)-one* (**4ao**). Yellow solid; Yield: 64%; Mp: 73.8–80.5 °C; IR (KBr plate) ν_max_ 2991, 2899, 1621, 1591, 1365, 825, 642. ^1^H NMR (600 MHz, CDCl_3_) δ 8.28–8.26 (d, *J* = 8.2 Hz, 1H, CH Ar), 8.14–8.12 (dd, *J* = 7.6, 4.8 Hz, 3H, CH Ar), 7.75–7.72 (m, 1H, CH Ar), 7.56–7.47 (dq, *J* = 30.1, 7.5 Hz, 4H, CH Ar), 7.18 (s, 1H, CH), 3.94 (s, 2H, CH_2_), 1.21 (s, 9H, CH_3_). ^13^C NMR (151 MHz, CDCl_3_) δ 162.42 (CO), 158.98, 149.17, 140.45, 129.96, 129.24, 129.17, 128.76, 127.62, 125.33, 121.69, 120.60, 98.65, 32.13, 26.77. HRMS Exact mass calcd. for C_20_H_22_ON [M + H]^+^: 292.16959; found: 292.16916.

### 3.3. Synthesis of Compounds **4ap**–**4ar**

General procedure. To a solution of DCE (1.5 mL) was added 2-aminoacetophenones **1a** (0.37 mmol), benzaldehyde **2a** (0.44 mmol), phenols **5** (0.37 mmol), PPA (0.37 mmol) and Pd/C (50 wt%, 19.6 mg, 5 mol% based on Pd content) in a 50 mL round bottom flask. The reaction mixture was refluxed for 3 h. The solution was quenched with water and the organic layer was dried over Na_2_SO_4_, filtered and evaporated. The resulting crude compound was purified by silica gel column chromatography using hexane/ethyl acetate mixtures to afford the corresponding products **4ap**–**4ar**.

*1,2-Diphenylquinolin-4(1H)-one* (**4ap**). White solid; Yield: 70%; Mp: 69.0–71.2 °C; IR (KBr plate) ν_max_ 1690, 1621, 1593, 1361, 835, 692. ^1^H NMR (600 MHz, CDCl_3_) δ 8.39–8.37 (d, *J* = 8.0 Hz, 1H, CH Ar), 8.20–8.19 (d, *J* = 8.5 Hz, 1H, CH Ar), 7.98–7,97 (d, *J* = 7.2 Hz, 2H, CH Ar), 7.81–7.78 (m, 1H, CH Ar), 7.60–7.57 (t, *J* = 7.6 Hz, 1H, CH Ar), 7.53–7.50 (t, *J* = 7.9 Hz, 2H, CH Ar), 7.48–7.42 (m, 3H, CH Ar), 7.35–7.33 (t, *J* = 7.5 Hz, 1H, CH Ar), 7.27–7.26 (d, *J* = 8.0 Hz, 2H, CH Ar), 7.06 (s, 1H, CH). ^13^C NMR (151 MHz, CDCl_3_) δ 162.39 (CO), 158.63, 154.64, 149.82, 139.85, 130.38, 130.34, 129.38, 129.33, 128.72, 127.51, 125.90, 125.50, 121.71, 120.96, 120.61, 102.57. HRMS Exact mass calcd. for C_21_H_16_ON [M + H]^+^: 298.12264; found: 298.12216.

*2-Phenyl-1-p-tolyl-quinolin-4(1H)-one* (**4aq**). White solid; Yield: 65%; Mp: 73.3–74.6 °C; IR (KBr plate) ν_max_ 1691, 1651, 1583, 1360, 845, 682. ^1^H NMR (600 MHz, CDCl_3_) δ 8.39–8.38 (m, 1H, CH Ar), 8.18–8.19 (d, *J* = 8.5 Hz, 1H, CH Ar), 7.98–7.96 (m, 2H, CH Ar), 7.80–7.79 (ddd, *J* = 8.4, 6.9, 1.5 Hz, 1H, CH Ar), 7.59–7.57 (ddd, *J* = 8.1, 6.9, 1.1 Hz, 1H, CH Ar), 7.48–7.41 (m, 3H, CH Ar), 7.31–7.30 (d, *J* = 8.1 Hz, 2H, CH Ar), 7.16–7.14 (m, 2H, CH Ar), 7.03 (s, 1H, CH), 2.45 (s, 3H, CH_3_). ^13^C NMR (151 MHz, CDCl_3_) δ 162.73 (CO), 158.66, 152.18, 149.74, 139.94, 135.24, 130.82, 130.33, 129.32, 129.28, 128.69, 127.54, 125.81, 121.74, 120.83, 120.56, 102.18, 20.94. HRMS Exact mass calcd. for C_22_H_18_ON [M + H]^+^: 312.13829; found: 312.13785.

*1-(4-Chlorophenyl)-2-phenylquinolin-4*(*1H*)-one (**4ar**). White solid; Yield: 47%; Mp: 93.9–112.9 °C; IR (KBr plate) ν_max_ 1696, 1631, 1591, 1366, 855, 672. ^1^H NMR (600 MHz, CDCl_3_) δ 8.34–8.32 (dd, *J* = 8.4, 1.5 Hz, 1H, CH Ar), 8.21–8.20 (d, *J* = 8.5 Hz, 1H, CH Ar), 7.99–7.97 (m, 2H, CH Ar), 7.82–7.99 (ddd, *J* = 8.4, 6.8, 1.4 Hz, 1H, CH Ar), 7.60–7.58 (m, 1H, CH Ar), 7.50–7.44 (m, 5H, CH Ar), 7.22–7.19 (m, 2H, CH Ar), 7.05 (s, 1H, CH). ^13^C NMR (151 MHz, CDCl_3_) δ 162.07 (CO), 158.58, 153.24, 130.77, 130.55, 130.43, 129.49, 129.40, 128.91, 128.79, 128.30, 127.50, 126.08, 122.26, 121.57, 120.44, 102.61. HRMS Exact mass calcd. for C_21_H_15_ONCl [M + H]^+^: 332.08367; found: 332.08319.

### 3.4. Biological Evaluation

#### 3.4.1. Cell Culture Conditions

Differentiated PC12 cells were cultured in DMEM supplemented with 2.5% fetal bovine serum and and 1% penicillin/streptomycin. Cells were cultured in a humidified atmosphere containing 5% CO_2_ at 37 °C.

#### 3.4.2. Protective Effects of 4-Quinolones 4a-4ar against NMDA-induced Injury in PC12 Cells

PC12 cells were plated on 96-well plates at a density of 4 × 10^3^ cells/well in 200 μL volumes. After 96 h of incubation, the cells were pretreated with MK-801 and 4-quinolones (20 µM or 0.5–30 µM) for 24 h. NMDA (2 mM) was then added for another 6 h to establish the cell injury model. After these treatments, 0.5 mg/mL MTT was added to the medium. After incubating for an additional 4 h, the medium was replaced by 100 µL DMSO. The absorbance was measured at 490 nm with a microplate reader (Thermo Scientific Varioskan LUX Multimode Reader, Waltham, MA, USA).

#### 3.4.3. Statistical Analysis

Data were expressed as mean ± SD. Multiple group differences were evaluated using one-way analysis of variance (ANOVA) followed by the post hoc LSD test. *p* < 0.05 were considered statistically significant.

## 4. Conclusions

In conclusion, we have developed a novel strategy for the synthesis of *N*-alkyl-4-quinolones via mechanistically intriguing three-component tandem reactions of 2-aminoacetophenones, aldehydes and alcohols. Our work provided the first proof of PPE as a dual hydrogen-bonding catalyst. Notable advantages of this protocol include readily accessible reagents, a one-pot procedure, broad substrate scope and being transition-metal free. The neuroprotective activity evaluation showed that compound **4h** demonstrated good protective potency.

## Data Availability

Not applicable.

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
