# Peer review of "Hydrogen Bond Assisted Three-Component Tandem Reactions to Access N-Alkyl-4-Quinolones"

_molecules, 2023, doi:10.3390/molecules28052304_

Round 1

Reviewer 1 Report

This is a good study (the methodology appeared to be new, the reaction mechanism proposed is reasonable, compound characterization is adequate and the biology data presented are interesting) and may be considered further after some minor revision / clarifications.

1.      Abstract: the last sentence should be started as “The compound 4h………….”

2.      Scheme 1: the driving force for the reaction of 6 with 9 is not clear.

3.      It is indicated that the presence of groups like F, Cl or Br decreased the product yield. However, I would like to know the effect of the genuine electron withdrawing group such as nitro or cyano. Similarly, the effect of a strong electron donating group such as OMe should be evaluated.

4.      Page 8, line 172-173 should be revised to “………was dried over Na2SO4 , filtered and evaporated.

5.      NMR data: it is desirable to assign all the characteristic peaks in both 1H and 13 C NMR data of each compound.

6.      Docking results: it is desirable to include the 2D interaction diagrams also for clear view and better understanding. Also include the distance of interaction at least for H-bonds.

7.      Fig S-2: what is the reason for selecting 20 uM concentration. Was MK-801 also used at the same concentration?  

8.      Though not essential but author could have presented some insight in terms of structure activity relationship based on the date of Fig S-2.

Reviewer 2 Report

The manuscript described a PPA-promoted cascade cyclization to synthesize 1-alkyl-2-aryl-quinolins from commercially available substrates, the method is concise and efficient with good generality, and the mechanism studies are reasonable at this stage. The overall scientific novelty and quality of this manuscript are sufficient to be published on Molecules after some minor issues are addressed.

1.     Describing the P2O5 and PPA as “catalyst” is inappropriate, since the optimal amount of them in reaction is 1.0 eq.

2.     All of the NMR spectra should include the information of “d-sovlent” (eg. CDCl3, 400MHz.)

3.     The decimal places of calculated and experimental HRMS should be the same (4f, line233; 4ad, line438;).

4.     The type of yield in Scheme 2 and 3 should be indicated.

Reviewer 3 Report

The article “Hydrogen Bond Assisted Three-component Tandem Reactions to Access N-Alkyl-4-quinolones” by Liu et al. describes one-pot versatile synthesis of N-substituted 4-quinolones. The authors described hydrogen bonding catalyst and possible reaction mechanism. Also, neuroprotective activity was investigated, compound 4h demonstrated good activity. The article is well written with negligible number of typos:

Typos at L. 131 and 133: electron-withdrawing instead of “electron-with-drawing”; at L. 174: hexane instead of “hexanes”; at L. 564: 4a-4ar should be in bold; at L. 566: on 96-well.

The manuscript requires minor revision.

Questions and comments related to the manuscript:

 1)     The compound 4h fit the active site with binding energy -7.46 kcal/mol, to better understand this value could you provide the binding energy of the co-crystallized ligand DCKA with NMDA receptor?

2)     In the section 3.2, why was a 50 ml flask used? The volume of reaction mixture does not exceed 2-3 ml.

3)     How did you evaluate the purity of the target compounds?
